# Dimethyl Bisphenolate Ameliorates Carbon Tetrachloride-Induced Liver Injury by Regulating Oxidative Stress-Related Genes

**DOI:** 10.3390/molecules28247989

**Published:** 2023-12-07

**Authors:** Rong Wang, Huanhuan Shen, Jiaxiang Zhang, Xiyan Li, Yang Guo, Zhenjun Zhao, Pingyu Wang, Ning Xie, Youjie Li, Guiwu Qu, Shuyang Xie

**Affiliations:** 1Department of Biochemistry and Molecular Biology, Binzhou Medical University, Yantai 264003, China; wr17616156627@163.com (R.W.); wazyshen2h@163.com (H.S.); zhangjiaxiang2580@163.com (J.Z.); 137070604@163.com (X.L.); guoyang010218@163.com (Y.G.); youjie1979@163.com (Y.L.); 2Shandong Laboratory of Advanced Materials and Green Manufacturing (Yantai), Yantai 264000, China; 3College of Life Sciences, Yantai University, Yantai 264005, China; zhaozhenjun@ytu.edu.cn; 4Department of Epidemiology, Binzhou Medical University, Yantai 264003, China; wpingyugirl@163.com; 5Department of Breast and Thyroid Surgery, Yantai Affiliated Hospital of Binzhou Medical University, Yantai 264000, China; sunnyvvv@163.com

**Keywords:** liver injury, dimethyl bisphenolate, biochemical indicators, Nrf2, oxidative stress

## Abstract

Liver disease accounts for millions of deaths per year all over the world due to complications from cirrhosis and liver injury. In this study, a novel compound, dimethyl bisphenolate (DMB), was synthesized to investigate its role in ameliorating carbon tetrachloride (CCl_4_)-induced liver injury through the regulation of oxidative stress-related genes. The structure of DMB was confirmed based on its hydrogen spectrum and mass spectrometry. DMB significantly reduced the high levels of ALT, AST, DBIL, TBIL, ALP, and LDH in a dose-dependent manner in the sera of CCl_4_-treated rats. The protective effects of DMB on biochemical indicators were similar to those of silymarin. The ROS fluorescence intensity increased in CCl_4_-treated cells but significantly weakened in DMB-treated cells compared with the controls. DMB significantly increased the content of oxidative stress-related GSH, Nrf2, and GCLC dose-dependently but reduced MDA levels in CCl_4_-treated cells or the liver tissues of CCl_4_-treated rats. Moreover, DMB treatment decreased the expression levels of P53 and Bax but increased those of Bcl2. In summary, DMB demonstrated protective effects on CCl_4_-induced liver injury by regulating oxidative stress-related genes.

## 1. Introduction

Hepatitis induces liver damage and liver cancer, which, together, cause a million people to die annually. The WHO warns that the diseases induced by viral hepatitis will kill more people than tuberculosis, malaria, and HIV combined by 2040 if the infection trends continue [1]. The Asia–Pacific region accounts for more than 60% of global deaths due to liver diseases. Chronic hepatitis B virus, alcohol consumption, and non-alcoholic fatty liver disease have caused more than half of the deaths attributed to cirrhosis in the region [2]. Many drugs have been deemed to be related to drug-induced liver injury, in which the anti-infective group of drugs is the most frequently implicated [3]. Drug-induced hepatocellular jaundice increases the chance of progressing to death or transplantation for patients [4]. Therefore, there must be a renewed focus on preventing liver injury and reducing the burden of liver diseases.

Drug toxicity can trigger a sterile inflammatory response to induce liver injury, which will cause phagocytes to produce reactive oxygen species (ROS) [5]. ROS-induced liver injury during inflammation is associated with the promotion of mitochondrial dysfunction through intracellular oxidant stress [6]. Khan et al. reported CCl_4_-induced hepatotoxic liver injuries in mice with the activation of the cytochrome pathway, the formation of ROS, and mitochondrial damage [7]. The generation of ROS can further trigger apoptotic and necrotic cell death.

Nuclear factor-erythroid 2-related factor 2 (Nrf2) is an important transcription factor in neutralizing cellular ROS and restoring redox balance [8]. Nrf2 can govern the expression levels of endogenous antioxidants and ROS-related genes in various electrophilic compounds [9]. Berberine exerts a protective effect against CCl_4_-induced acute liver injury by effectively regulating the expression of Nrf2-Keap1-related genes and suppressing p53-mediated hepatocyte apoptosis [10]. Various phenolic compounds can cause nuclear factor-κB or Nrf-2 to play important roles in antioxidant responses [11]. Phenolic compounds have anti-inflammatory properties in the treatment of rheumatoid arthritis or inflammatory bowel disease [11].

In a previous study, we found that bisphenolate-ND-309 had antioxidant properties, and it significantly increased the ATP content, attenuated the elevation of malondialdehyde (MDA) levels, and decreased the generation of ROS [12]. This study aimed to identify a new compound that ameliorates liver injury and to study its mechanism of antioxidation. A more active compound (DMB) was synthesized on the basis of bisphenol acid and the previous study. The results demonstrate that it has protective effects on CCl_4_-induced liver injuries.

## 2. Results

### 2.1. Synthesis and Structure of DMB

After the reaction, a total of 10.4 g of yellow powder was obtained through purification via HPLC, with a yield rate of 96.3% and a purity of 99.3%.

The NMR analysis results of DMB is as follows: ^1^H NMR (600 MHz, *d*_6_-DMSO) δ 9.97 (s, 1H, -OH), 9.53 (s, 1H, -OH), 8.84 (s, 1H, -OH), 8.49 (s, 1H, -OH), 7.73 (s, 1H), 7.34 (d, *J* = 15.9 Hz, 1H), 7.30 (d, *J* = 8.4 Hz, 1H), 6.88 (d, *J* = 8.4 Hz, 1H), 6.56 (d, *J* = 8.2 Hz, 1H), 6.46 (dd, *J* = 8.6, 2.2 Hz, 1H), 6.44 (d, *J* = 2.1 Hz, 1H), 6.28 (d, *J* = 15.9 Hz, 1H), 3.63 (s, 3H), 3.60 (s, 3H). 

^13^C NMR (151 MHz, *d*_6_-DMSO) δ 168.00, 167.40, 148.12, 148.08, 145.40, 143.55, 143.04, 142.26, 126.21, 125.87, 124.24, 123.72, 122.64, 119.01, 117.88, 115.78, 115.76, 115.28, 52.38, 51.66.

The high-resolution mass spectrometry analysis results were as follows: chemical formula: C_20_H_18_O_8_; exact mass: 386.10; molecular weight: 386.36; *m*/*z*: 386.10 (100.0%), 387.10 (21.6%), 388.11 (2.2%), 388.10 (1.6%); elemental analysis: C, 62.18; H, 4.70; O, 33.13 (Figure 1A).

The structure of DMB was confirmed on the basis of the above hydrogen spectrum and mass spectrometry data, combined with the structural characteristics of the reactant (bisphenol acid and methanol), as shown in Figure 1B.

### 2.2. Biochemical and Histological Changes in CCl_4_-Induced Liver Injury through Oxidative Stress

CCl_4_ is commonly used in establishing acute liver injury models through the modulation of oxidative imbalance [13,14]. On the basis of the concentration of CCl_4_ in a previous report [13], five different injection doses, namely, 0, 300, 600, 900, and 1200 μL/kg, were designed to treat SD rats to further determine the appropriate dosage needed in this study. One day after a single injection, the blood of the rats was collected from the apex of the heart, and the sera were tested.

The alanine aminotransferase (ALT), aspartate aminotransferase (AST), direct bilirubin (DBIL), total bilirubin (TBIL), alkaline phosphatase (ALP), and lactate dehydrogenase (LDH) in the sera are the most common liver function indices to estimate liver injury [15,16,17]. When the dose of CCl_4_ was more than 300 μL/kg, the levels of ALT (Figure 2A) and AST (Figure 2B) in the sera of the rats significantly increased in a dose-dependent manner. The levels of ALT and AST reached the highest under 1200 μL/kg CCl_4_ treatment. DBIL began to increase under a dose of 600 μL/kg of CCl_4_ in a dose-dependent manner, and it reached the maximum at 1200 μL/kg CCl_4_ (Figure 2C). Serum TBIL increased significantly up to 900 μL/kg of CCl_4_ injections (Figure 2D). ALP and LDH increased significantly with an injection of 600 μL/kg and 1200 μL/kg CCl_4_, respectively (Figure 2E,F).

Histological changes in the liver structure are important to estimate liver injury [18]. HE staining demonstrated that the liver cell cords of the control rats without an injection of CCl_4_ were arranged neatly and regularly, with no infiltration of inflammatory factors. A small number of inflammatory cells infiltrated in the liver tissues of the rats after 300 μL/kg CCl_4_ treatment, but the liver cords remained neat and regular. The liver tissues of the rats treated with 600 and 900 μL/kg CCl_4_ showed significant vacuolar lesions, pyknosis of the nucleus, and extensive infiltration of inflammatory cells in a dose-dependent manner. After 1200 μL/kg of CCl_4_ treatment, most cells were on the verge of necrosis, and they demonstrated severe liver damage (Figure 2G). The results indicate that CCl_4_ dose-dependently has a significant effect on liver injury.

On the basis of the above results, significant biochemical and histological changes were found in the 900 μL/kg CCl_4_-treated rats. Therefore, 900 μL/kg CCl_4_ was selected for the following study of the effects of DMB.

### 2.3. DMB Ameliorated Biochemical and Histological Changes in CCl_4_-Induced Liver Injury

The SD rats were administered with different concentrations of DMB (low dose, 15 mg/Kg; medium dose, 30 mg/Kg; and high dose, 45 mg/Kg) via gastric gavage for 1 week to investigate the protective roles of DMB in liver injury. The rats were then treated with 900 μL/Kg CCl_4_. The results demonstrate that DMB significantly reduced the high levels of ALT (Figure 3A) and AST (Figure 3B) in a dose-dependent manner in the sera of the CCl_4_-induced liver injury rats. DMB also reduced the upregulating trends of DBIL (Figure 3C), TBIL (Figure 3D), ALP (Figure 3E), and LDH (Figure 3F) caused by CCl_4_. The protective effects of DMB on the biochemical indicators were similar to or better than those of 43.75 mg/Kg silymarin (clinical drug used as positive control treatment, Figure 3A–F).

HE staining further supported the notion that the low, medium, and high doses of DMB ameliorated the histological liver injury induced by CCl_4_, similar to silymarin (Figure 3G). These results indicate that DMB can effectively reduce the indicators of CCl_4_-induced liver injury.

### 2.4. DMB Increased Oxidative Stress-Related Nrf2/GCLC in CCl_4_-Induced Liver Injury of Rats

Emerging evidence supports a link between oxidative stress and mitochondrial dysfunction in CCl_4_-induced liver injury [13,19]. The above-mentioned results show that DMB ameliorated CCl_4_-induced liver injury. The possible binding proteins were predicted using SEA and SuperPred to further investigate the molecular mechanism of DMB in liver injury, and the results demonstrate that Nrf2 is a target of DMB (Appendix A). Moreover, arginine (ARG) at position 322, valine (VAL) at position 420, glycine (GLY) at position 511, and VAL at position 512 in NRF2 can bind to DMB through hydrogen bonds (Figure 4A).

These results indicate that Nrf2 is a target of DMB. Nrf2 functions as a ubiquitous and evolutionarily conserved protein to counteract oxidative stress [20] and, thus, participates in the pathological process of liver injury [21]. Therefore, the roles of Nrf2 in liver injury were investigated in this study. Nrf2, as a controller of enzymes, is important for producing glutathione (GSH) and MDA [22]. The levels of GSH and MDA were detected to study the roles of DMB-related Nrf2 in CCl_4_-induced liver injury. The results show that CCl_4_ reduced the content of GSH in the liver tissues compared with the control. DMB treatment significantly increased the content of GSH in the liver tissues of the CCl_4_-treated rats (Figure 4B). Meanwhile, CCl_4_ significantly increased the content of MDA in the liver tissues, and this content was significantly reduced after DMB treatment. As the concentration of DMB increased, the content of MDA gradually decreased (Figure 4C). Similar results of GSH and MDA were found in the silymarin-treated rats (Figure 4B,C). Compared with the normal controls, the expression levels of Nrf2 and GCLC decreased in the liver tissues of the CCl_4_-treated rats, and DMB treatment increased these levels in a dose-dependent manner (Figure 4D–F).

Nrf2 can inhibit p53 expression in the liver and cultured hepatocytes [23]. ROS, which is generated during oxidative stress, can enhance p53 expression and activate pro-oxidant genes to result in p53-dependent apoptosis [24]. The expression levels of P53, Bax, and Bcl2 in the liver tissues were detected. The results show that the expression of P53 and Bax increased in the liver tissues of CCl_4_-induced models, whereas that of Bcl2 decreased compared with the controls. Meanwhile, DMB treatment decreased the expression levels of p53 and Bax but increased those of Bcl2, and the changes in p53, Bax, or Bcl2 induced by DMB were similar to the effects of silymarin on the expression of these genes (Figure 4D,G,H). The results further indicate that the molecular mechanism of DMB in protecting against liver injury is related to the regulation of the expression levels of P53, Bax, and Bcl2.

### 2.5. DMB Ameliorated Oxidative Stress Induced by CCl_4_ to HepG2 and MHCC-97H Cells

The above-mentioned results demonstrate that DMB has a protective effect on liver injury in rats. The roles of DMB in ameliorating the injury induced by CCl_4_ in vitro were further investigated in HepG2 and MHCC-97H cells. First, 10 different volume concentrations were set. Second, the half maximal inhibitory concentration (IC_50_) of CCl_4_ on HepG2 and MHCC-97H cells was determined through CCK-8 experiments. The IC_50_ values in HepG2 and MHCC-97H cells were determined to be 0.3% (Figure 5A) and 0.25% (Figure 5B), respectively.

A 3-(4,5-Dimethylthiazol-2-yl)-2,5-diphenyltetrazolium bromide (MTT) experiment further indicated that DMB effectively protected the cells from the injury induced by CCl_4_ in vitro. When the concentration of DMB reached 20 and 30 μg/mL, the injury induced by CCl_4_ to the cells was almost fully reversed, and the cell activity was similar to that after control treatment (Figure 5C,D). Therefore, 20 μg/mL was selected as the experimental concentration of DMB in the following experiments.

Then, the MDA and GSH levels were detected to study the roles of DMB in reversing CCl_4_-induced cell injury in vitro. The results show that the increase in MDA levels or the decrease in GSH levels induced by CCl_4_ was reversed in DMB-treated HepG2 or MHCC-97H cells compared with that in the controls (Figure 5E,F), similar to the results of the in vivo study.

Oxidative stress is a phenomenon of ROS accumulation in cells [25]. An increase in GSH can reduce the contents of ROS and MDA to ameliorate oxidative stress [26]. The results of the present study show that the ROS fluorescence intensity significantly increased in CCl_4_-treated HepG2 or MHCC-97H cells and significantly weakened in DMB-treated cells compared with the controls (Figure 5G,H). The results demonstrate that DMB has antioxidant effects on reducing MDA and ROS by increasing GSH levels in vitro.

Western blot and immunofluorescence analyses further demonstrated that CCl_4_ treatment reduced the expression levels of Nrf2 and GCLC in HepG2 or MHCC-97H cells compared with those in the controls (Figure 6A–D). Nevertheless, DMB ameliorated the roles of CCl_4_-reduced Nrf2 and GCLC in cells (Figure 6A,B), indicating that the antioxidant effects of DMB are related to the regulation of Nrf2 and GCLC expression levels.

### 2.6. DMB Reversed Cell Apoptosis Induced by CCl_4_

The results demonstrate that CCl_4_ treatment significantly increased the number of apoptotic cells compared with the control treatment. DMB ameliorated the effects of CCl_4_-induced cell apoptosis in HepG2 and MHCC-97H cells (Figure 7A,B). Western blot further revealed that CCl_4_ treatment increased p53 and Bax expression and decreased Bcl-2 levels, whereas DMB reversed these changes (Figure 7C,D). The results further support the notion that the anti-apoptotic role of DMB is related to the regulation of the expression of p53-related genes.

The above-mentioned results indicate that DMB ameliorates the roles of CCl_4_-induced liver injury by regulating Nrf2-related genes. Next, the cytotoxicity of DMB in vitro and in vivo was assessed. The results show that 10, 20, 30, 60, and 100 μg/mL DMB treatments were not cytotoxic to MHCC-97H and 293T cells (Appendix A). In the in vitro study, the treatment of 45 mg/kg DMB did not induce significant changes in the liver tissues (Appendix A) according to a histological analysis. The results support the notion that DMB does not significantly induce cytotoxicity in vitro and in vivo.

## 3. Discussion

Liver disease, which is a significant clinical problem, accounts for millions of deaths per year all over the world due to complications from cirrhosis and liver injury [27]. Thus, new treatment for liver injury is urgently needed. A biochemical analysis demonstrated that lachnum polysaccharide treatment improved the antioxidant status (GSH and MDA) and showed protection against liver injury [28]. Salvia miltiorrhiza polysaccharides demonstrated a good protective effect on liver damage by increasing the content of GSH and decreasing the levels of ALT, AST, and MDA [29]. In this study, a new compound (DMB) was synthesized, and the results show that DMB ameliorates the roles of CCl_4_-induced liver injury by increasing Nrf2, GCLC, and GSH levels and decreasing ROS levels (Figure 8).

CCl_4_ is commonly used to induce liver injury in animal models. An intraperitoneal injection of CCl4 can result in liver inflammation and oxidative stress through the downregulation of superoxide dismutase and GSH [30]. CCl_4_ increased plasma ALT and AST activities and induced hepatic fibrosis formation by decreasing α-SMA and collagen proteins [31]. We found that significant changes in ALT, AST, DBIL, TBIL, and liver structures occurred in the CCl_4_-treated rats; these results are supported by those of Ebaid [32], who showed that CCl_4_ significantly upregulated the levels of lipid peroxidation (MDA), cholesterol, LDL, and bilirubin but significantly suppressed the activity of GSH.

GSH, which is a ubiquitous intracellular peptide, plays important roles in detoxification and antioxidant defense [33]. GSH synthesis is regulated by the activity of the rate-limiting enzyme glutamate cysteine ligase (GCL), which is composed of a catalytic (GCLC) and a modifier subunit [34]. Oxidative stress can induce the expression of GSH synthetic enzymes to further affect ROS levels. The Nrf2 pathway, a powerful sensor for the cellular redox state, is activated directly by oxidative stress, and it can increase GCL levels and the content of GSH [35]. Chen et al. showed that the nuclear translocation of Nrf2 increased in 5-caffeoylquinic acid (5-CQA)-treated HepG2 cells. 5-CQA can significantly increase the Nrf2-antioxidant response element activity, and Nrf2 inhibition completely antagonized the ability of 5-CQA to induce GCL expression [36]. Here, we found that the new compound DMB significantly increased the content of GSH, Nrf2, and GCLC in the liver tissues of the rats with CCL_4_-induced liver injury. Considering the roles of ROS in inflammation [25], our results indicate that DMB may ameliorate liver inflammation by regulating GSH, Nrf2, and GCLC to reduce ROS.

Nrf2 can affect the expression of p53 in the liver and cultured hepatocytes [23]. Moreover, the livers of the CCl_4_-treated mice exhibited a marked increase in the expression of P53 and Bax and a downregulation in Bcl2 expression [37]. The results in this study show similar changes in the p53-related genes of CCl_4_-induced liver injury tissues. Western blot showed that the p53 and Bax expression levels increased and that the Nrf2 and Bcl2 levels decreased in CCl_4_-induced liver cells or tissues. Meanwhile, DMB treatment decreased the expressions of P53 and Bax but increased those of Nrf2 and Bcl2 compared with CCl_4_-treated cells or tissues.

Silymarin is a hepatoprotective agent widely used for the therapy of liver injury [38]. It acts as an antioxidant that can reduce free radical production and lipid peroxidation, and it has antifibrotic activity. Considering its protective roles in liver injury, silymarin was used to ameliorate liver injury caused by CCl_4_, acetaminophen, phenylhydrazine, and alcohol [39]. Hussein et al. showed that silymarin protected against liver injury via antioxidant and anti-apoptotic activities through the regulation of Keap1-Nrf2/HO-1 expression [40]. Considering its roles in protecting liver injury through antioxidant and anti-apoptotic activities, silymarin was used as a positive control in this study. Interestingly, the results indicate that the protective effects of DMB on ameliorating liver injury were similar to or better than those of silymarin. Therefore, DMB has potential application or combination with silymarin in the protection against liver injury.

To sum up, our results show that CCl_4_ can induce liver injury through two aspects. It increases oxidative stress by suppressing the expression of Nrf2 and GCLC to reduce GSH levels. CCl_4_ increases ROS levels to further induce cell apoptosis by regulating p53-related genes. The results show that DMB ameliorates the roles of CCl_4_-induced liver injury. The possible mechanism may be related to the increase in Nrf2, GCLC, and GSH levels and the decrease in ROS. DMB may be applied in clinical practice for liver injury in the future. However, the detailed regulating mechanism and the cytotoxicity of DMB, such as its side effects and concentration in vivo, must be further investigated.

## 4. Materials and Methods

### 4.1. DMB Synthesis

Bisphenol acid (10.0 g, Yantai Institute of Chemical Industry, Yantai, China) was dissolved in 100 mL methanol, and then hydrochloric acid was added to react for 4 h. A total of 10.4 g yellow powder was obtained after reducing the pressure at 50 °C and drying at vacuum. The yellow powder was mixed with silica gel at a mass ratio of 1:1 and eluted with a mixture gradient of chloroform and methanol. Toluene-trichloromethane-ethyl-acetate-methanol formic acid (ratio, 2:3:4:0.5:0.2) was detected, and the highest content of the eluted portion was collected to dry at 50 °C. Then, HPLC was used to analyze the DMB as follows: C18 reverse-phase chromatographic column, detection wavelength of 286 nm, and mobile phase of 2% formic acid (A)–acetonitrile (B). The gradient elution procedure was as follows: 0−15 min, 10% B to 100% B; 15−20 min, 100% B; 20−25 min, 100% B to 10% B; 25−30 min, 10% B.

### 4.2. DMB Structure Analysis

The DMB structure was analyzed using a nuclear magnetic resonance instrument (Advance III 50, Bruker, Billerica, MA, USA) and mass spectrometry (Q-Exactive, ThermoFisher, Norristown, PA, USA). The conditions of mass spectrometry were as follows: The ion source was HESI, with negative ion detection mode; the sheath gas flow rate was 35 arb. The auxiliary gas volume flow rate was 10 arb; the spray voltage was 2.80 kV. The temperature of the ion transfer tube and auxiliary gas was 325 °C and 350 °C, respectively. The scanning mode was as follows: Full MS/dd MS2; the resolution of Full MS was 70,000; the resolution of dd-MS2 was 17,500; and the scanning range was 100 to 1000 *m*/*z*.

### 4.3. Rat Model

SD rats (Jinan Pengyue Experimental Animal Breeding Co., Ltd., Jinan, China) were randomly divided by a “blinded” investigator into a normal control group; a model group; a silymarin group (43.75 mg/kg based on clinical medication conversion); and DMB low- (15 mg/kg), medium- (30 mg/kg), and high-dose groups (45 mg/kg), with 6 rats in each group. The drugs were administered at appropriate concentrations by using a 1% sodium carboxymethyl cellulose solution as the solvent. Each medication group was administered with a volume of 1 mL/300 g via gavage, once a day for 7 consecutive days. The normal control rats were given an equal volume of 1% carboxymethyl cellulose sodium solution via gavage. Two hours after the end of the previous administration, the model group and each drug group were given a one-time intraperitoneal injection of 900 μL/kg CCl_4_ (Fuyu Fine Chemical Co., Ltd., Tianjin, China) solution. The normal controls were intraperitoneally injected with an equal volume of olive oil. All animal experiments were approved by the Animal Ethics Committee of Binzhou Medical University (No. 2021-06-03).

### 4.4. Biochemical Indicator Analysis

Twenty-four hours after the intraperitoneal injection of the CCl_4_ solution into the rats, 2% pentobarbital sodium was administered intraperitoneally as anesthesia. Blood was taken from the rat’s apex by using a syringe and placed at room temperature for 2 h. The ALT, AST, DBIL, TBIL, ALP, and LDH levels in the sera were analyzed by using an Advia2400 biochemical analyzer (Siemens, Munich, Germany). The detailed detection of these indicators is shown in the supplemental methods.

### 4.5. Pathological Changes in Liver Tissues

The liver tissues were separated and fixed in 4% paraformaldehyde solution (BL539A, biosharp, Shanghai, China) for 24 h. The tissues were dehydrated, embedded in paraffin, sliced to a thickness of 4 μm, and stained with HE. The pathological changes in the rat liver tissues were observed under a microscope (DM6000B, Leica, Dresden, Germany) by a liver pathologist.

### 4.6. Detection of MDA and GSH Content

The frozen liver tissues were thawed at room temperature. Following the instructions of the corresponding MDA reagent kit (S0131S, Beyotime Biotechnology, Shanghai, China), 0.1 mL samples were added into each vial with 0.2 mL MDA detection working solution. The mixture was mixed and incubated in boiling water. Then, it was cooled and centrifuged at 1000× *g* for 10 min at room temperature. ELISA was used to determine the contents of MDA in the liver tissues.

GSH was analyzed using a GSH kit in accordance with the manufacturer’s instructions (S0053, Beyotime Biotechnology, Shanghai, China). Briefly, reactions, including samples, protein removal reagent M solution, and total glutathione assay working solution, were prepared in 96-well plates and incubated at room temperature for 5 min. Then, 50 µL of 0.5 mg/mL NADPH solution was added to each well, mixed, and detected using ELISA.

### 4.7. Prediction of DMB-Related Proteins

The structure of DMB (SMILES) was identified using SwissTargetPrediction (http://swisstargetprediction.ch/ (accessed on 16 October 2022)) to predict DMB-related proteins. Based on the structure of DMB, the possible binding proteins were predicted using SEA (http://sea.bkslab.org/ (accessed on 16 October 2022)) and SuperPred (https://prediction.charite.de/index.php?site=chemdoodle_search_target (accessed on 16 October 2022)).

The structure of the Nrf2 protein was downloaded through RCSB PDB, and CHem 3D 20.0 software (CJMarketing, Suzhou, China) was used to convert the secondary structure of DMB into a 3D structure for optimization. PyMOL 2.6 software (DeLano Scientific LLC, USA) was used to remove water and residue from Nrf2, and AutoDockTools 4.26 software (Scripps Research Institute, USA) was used for molecular simulation docking.

### 4.8. Cell Culture

HepG2 and MHCC-97H cells were obtained from the Shanghai Institute of Cell Biology, China. They were cultured in a DEME medium (high glucose; Hyclone, Logan, UT, USA) supplemented with 10% FBS and 5% CO_2_ at 37 °C.

After the HepG2 and MHCC-97H cells (5 × 10^3^ cells each well) were inoculated on 96-well plates for 24 h, different volume concentrations of CCl_4_ (from 0.05% to 0.4%) were added for another 24 h. After the culture medium was removed, CCK-8 (BS350C, Biosharp, China) was added to the cells and incubated at 37 °C for 2 h. The optical density (OD) was measured at 450 nm by using an ELISA reader (Multiskan FC, Thermo Fisher Scientific, Waltham, MA, USA) to calculate IC_50_.

### 4.9. MTT Assay

An MTT assay was used to analyze cell proliferation according to previous reports [41,42]. In brief, cells (1 × 10^4^) were cultured in each well of the 96-well plates. At 4 h before the end of incubation, 10 μL MTT (Sigma, St Louis, MO, USA, 5 mg/mL) was added to each well. After the supernatant was removed, 100 μL DMSO (Sigma) was added, and the OD was measured using an ELISA reader (Multiskan FC, Thermo Fisher Scientific).

### 4.10. ROS Content Analysis

The cells (5 × 10^6^) were inoculated in a six-well plate for 24 h, and the CCl_4_-treated cells were further incubated with or without DMB. They were then collected and stained using 10 μM DCFH-DA (S0033S, Beyotime Biotechnology, Shanghai, China) at 37 °C for 20 min. The fluorescence intensity was analyzed using a microscope (LEICA TCS SPE, Leica, Dresden, Germany).

### 4.11. Cell Apoptosis

Apoptotic cells were analyzed using an Annexin V-FITC/PI kit (KeyGEN Biotech, Co., Ltd., Nanjing, China). The cells were collected and resuspended in PBS gently. Then, they were centrifuged and resuspended in 195 μL Annexin V-FITC binding buffer. Annexin V-FITC (5 µL) was added, and 10 µL propidium iodide was mixed gently. The mixture was incubated in the dark for 10–20 min and examined immediately using flow cytometry (Beckman Coulter, Lane Cove West, NSW, Australia) as previously described [42,43].

### 4.12. Western Blot

The total proteins were extracted from the liver tissues or cells as previously described [42,44]; separated with 10% SDS-PAGE; and transferred to PVDF membranes, which were then blocked with 5% milk and incubated with primary and goat anti-rabbit IgG (H+L) secondary antibodies (1:6000, MB001, Bioworld, Minneapolis, MN, USA) separately. The membranes were analyzed using enhanced chemiluminescence substrates (P0018M, BeyoECL Plus, Beyotime, Nantong, China). The antibodies were as follows: rabbit anti-human or rat Nrf2 (1:2000, BS6286, Bioworld Technology, Ltd., Minneapolis, MN, USA), rabbit anti-human or rat glutamate cysteine ligase catalytic (GCLC, 1:6000, 12601-1-AP, Proteintech, Rosemont, IL, USA), rabbit anti-human or rat P53 (1:2000, 21891-1-AP, Proteintech), rabbit anti-human or rat Bax (1:1000, BS6420, Bioworld), rabbit anti-human or rat Bcl2 (1:1000, BS70205, Bioworld), and rabbit anti-human or rat GAPDH (1:3000, AP0063, Bioworld).

### 4.13. Immunofluorescence Analysis

Cell slides were washed with PBS and fixed at room temperature with 4% paraformaldehyde (BL539A, Biosharp, China) for 2 h. They were then broken using 0.2% TritonX-100 at room temperature for 15 min and sealed with a sealing solution at room temperature for 1 h. The first antibody rabbit anti-human Nrf2 (1:100, Bioworld Technology, Ltd., MN, USA) was incubated overnight, and the second antibody was incubated at 37 °C in the dark for 2 h and then incubated with DAPI (C1006, Beyotime Biotechnology, Shanghai, China) for 5 min. The fluorescence was analyzed using a microscope (DM6000 B, Leica, Dresden, Germany) as previously described [44].

### 4.14. Statistics

SPSS 22.0 software was used to analyze statistical significance (IBM Corp., Armonk, NY, USA). The median (interquartile range) is presented for abnormally distributed data, and multiple groups were compared using the Kruskal–Wallis H test. Normally distributed data are presented as mean ± SD. Multiple groups were compared using ANOVA. Tukey’s test was used for multiple comparisons. *p* < 0.05 indicated a statistically significant difference.

## 5. Conclusions

The novel compound DMB was synthesized, and it demonstrated similar protective effects on CCl_4_-induced liver injury to silymarin. The protective mechanism of DMB in liver injury is related to the regulation of the expression of Nrf2-related genes. This study shows the potential application of DMB to liver injury.

## Figures and Tables

**Figure 1 molecules-28-07989-f001:**
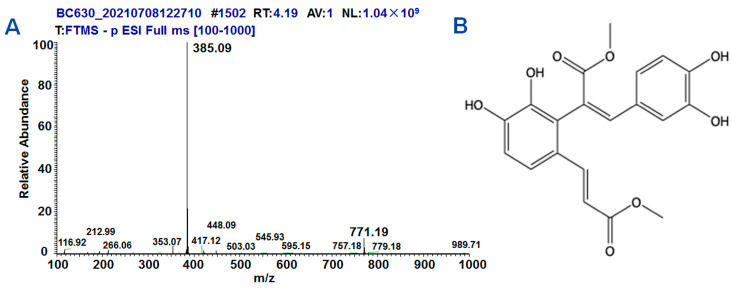
The synthesis and structure of DMB. (**A**) The results of high-resolution mass spectrometry. (**B**) The structure of DMB.

**Figure 2 molecules-28-07989-f002:**
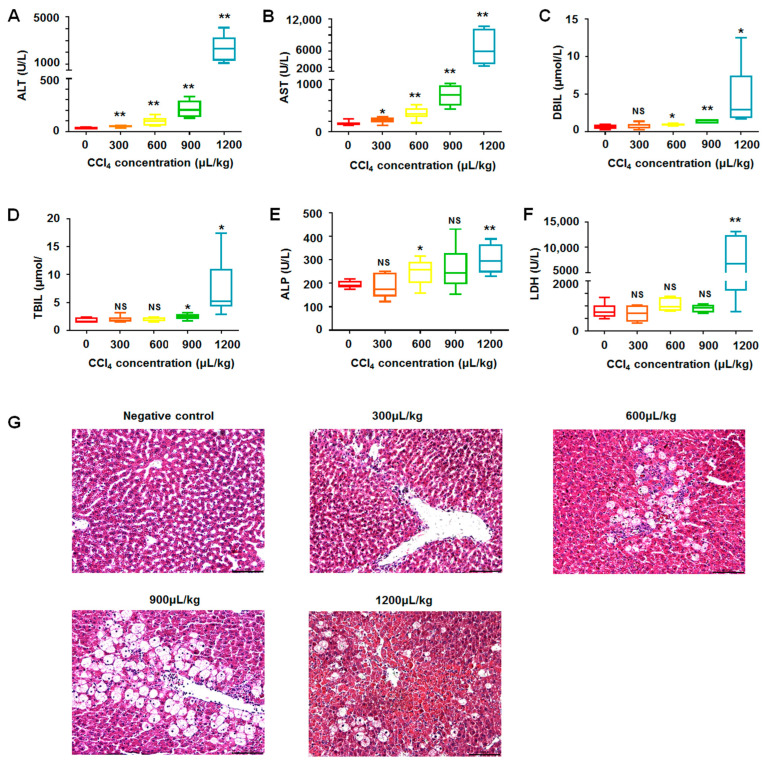
**The biochemical and histological changes in model rats.** (**A**) The levels of ALT in sera significantly increased in CCl_4_-induced models; (**B**) the levels of AST in sera significantly increased in CCl_4_-induced models; (**C**) DBIL changed significantly in sera of model rats. (**D**) TBIL increased significantly in sera of model rats. (**E**) The changes in ALP in the sera of model rats. (**F**) The changes in LDH in the sera of model rats. (**G**) The histological changes in liver tissues of model rats. Data were abnormally distributed, and they are expressed as median (interquartile range), * *p* < 0.05, ** *p* < 0.01; Kruskal–Wallis H test. NS, no statistical difference.

**Figure 3 molecules-28-07989-f003:**
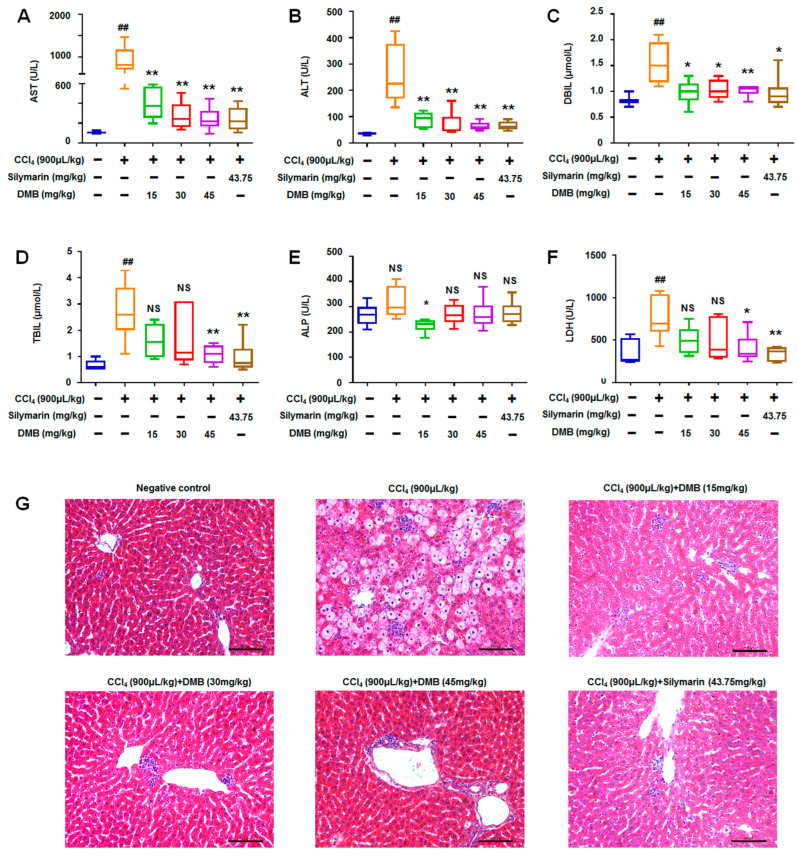
**DMB ameliorated the biochemical and histological changes in CCl_4_-induced models.** (**A**) DMB significantly decreased ALT levels in sera of CCl_4_-induced models; (**B**) the levels of AST were reduced in DMB+CCl_4_-treated rats; (**C**) DBIL decreased in sera of DMB+CCl_4_-treated models. (**D**) TBIL decreased in sera of DMB+CCl_4_-treated models. (**E**) The changes in ALP in the sera of DMB+CCl_4_-treated rats. (**F**) The changes in LDH in the sera of DMB+CCl4-treated rats. (**G**) DMB ameliorated the histological changes in liver tissues of CCl_4_-induced models. Data were abnormally distributed, and they are expressed as median (interquartile range), * *p* < 0.05, ** *p* < 0.01 vs. CCl_4_ treatment; ##, ** *p* < 0.01 vs. negative control; Kruskal–Wallis H test. NS, no statistical difference.

**Figure 4 molecules-28-07989-f004:**
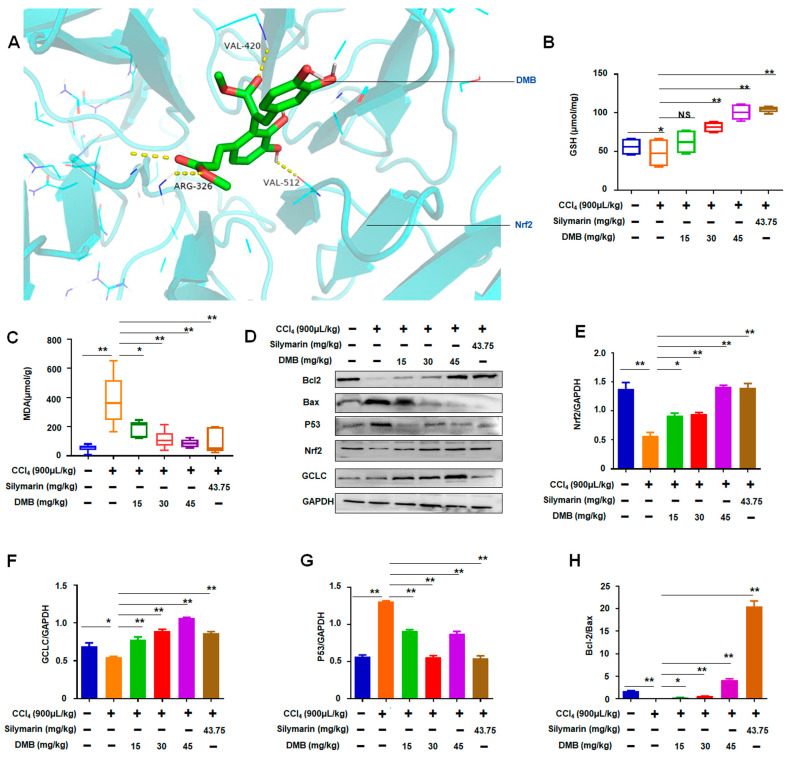
**DMB ameliorated the Nrf2-related factors in CCl_4_-induced liver injury.** (**A**) The related protein Nrf2 of DMB was predicted; (**B**) DMB increased the content of GSH in the liver tissues of CCl_4_-treated rats. Data were abnormally distributed, and they are expressed as median (interquartile range), * *p* < 0.05, ** *p* < 0.01; Kruskal–Wallis H test. (**C**) DMB reduced the MDA levels in the liver tissues of CCl_4_-treated rats. Data are expressed as median (interquartile range), * *p* < 0.05, ** *p* < 0.01; Kruskal–Wallis H test. (**D**) Western blot analysis of Nrf2-related gene expression in the liver tissues; (**E**) Relative GCLC expression in the liver tissues. Data are presented as the mean ± SD. * *p* < 0.05; ** *p* < 0.01; ANOVA. (**F**) Relative Nrf2 expression in the liver tissues. Data are presented as the mean ± SD. * *p* < 0.05; ** *p* < 0.01; ANOVA. (**G**) Relative p53 expression in the liver tissues. Data are presented as the mean ± SD. ** *p* < 0.01; ANOVA. (**H**) DMB decreased the expression of Bax but increased Bcl2 expression in the liver tissues. Data are presented as the mean ± SD. * *p* < 0.05; ** *p* < 0.01; NS, no statistical difference.

**Figure 5 molecules-28-07989-f005:**
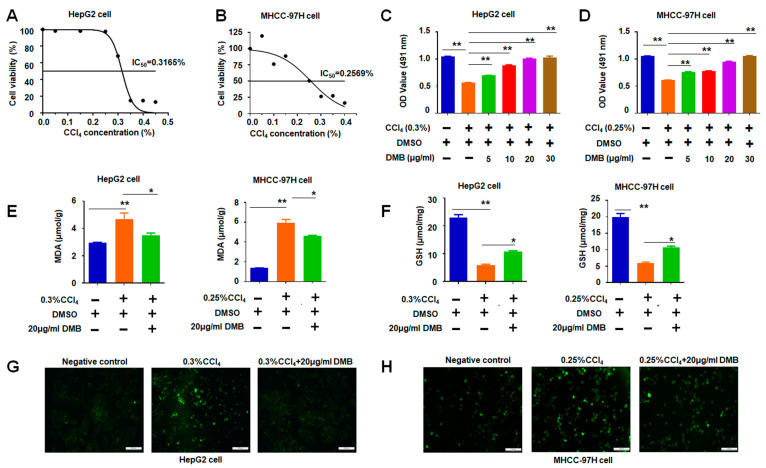
**DMB ameliorated the oxidative stress in vitro.** (**A**,**B**) The IC_50_ values of HepG2 cells and MHCC-97H cells were determined using CCK-8 method. (**C**,**D**) After treatment of CCl_4_ or different concentrations of DMB for 24 h, the activities of HepG2 or MHCC-97H cells were analyzed using MTT, respectively. Data are presented as the mean ± SD. ** *p* < 0.01; ANOVA. (**E**) The content of MDA in HepG2 and MHCC-97H cells was estimated using a microplate reader. Data are presented as the mean ± SD. * *p* < 0.05; ** *p* < 0.01; ANOVA. (**F**) GSH content in HepG2 and MHCC-97H cells was determined. Data are presented as the mean ± SD. * *p* < 0.05; ** *p* < 0.01; ANOVA. (**G**,**H**) ROS content in HepG2 and MHCC-97H cells was analyzed after CCl_4_ or MB intervention for 24 h.

**Figure 6 molecules-28-07989-f006:**
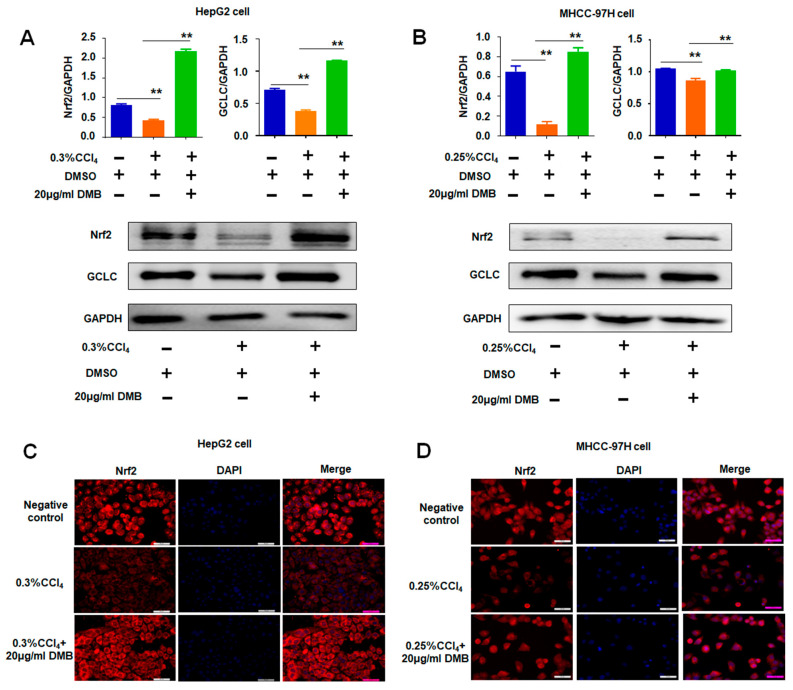
**DMB reversed CCl_4_-induced Nrf2 and GCLC expression.** (**A**,**B**) Western blot detected the expression of Nrf2 and GCLC in HepG2 and MHCC-97H cells, respectively. Data are presented as the mean ± SD. ** *p* < 0.01; ANOVA. (**C**,**D**) Immunofluorescence analyses of the expression of Nrf2 in HepG2 and MHCC-97H cells, respectively. Bar = 50 μm.

**Figure 7 molecules-28-07989-f007:**
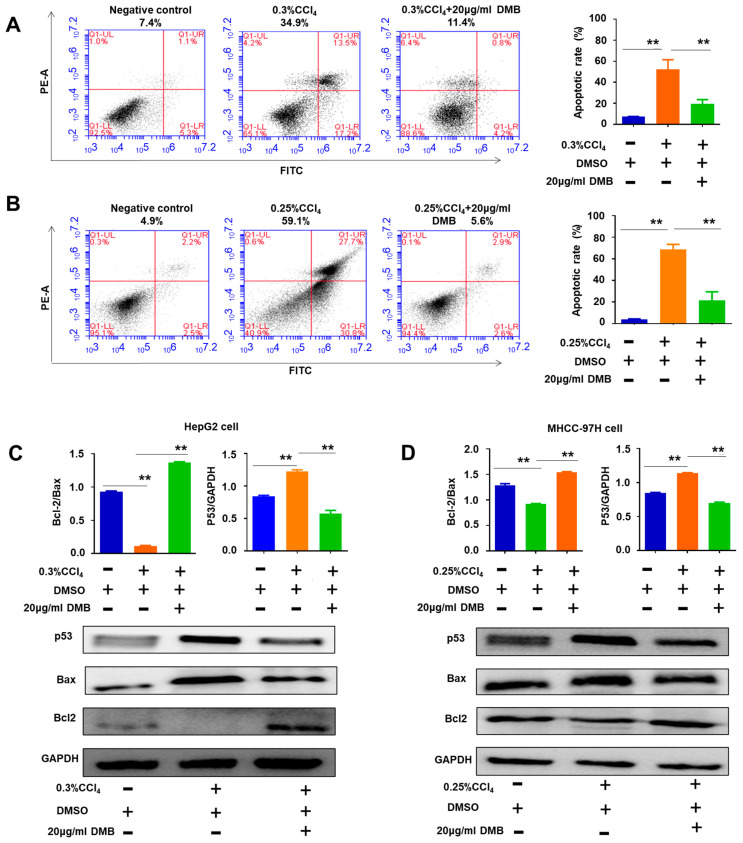
**DMB inhibited the apoptosis induced by CCl_4_.** (**A**,**B**) The apoptotic rates of HepG2 and MHCC-97H cells were measured using flow cytometry and statistically counted, respectively. Data are presented as the mean ± SD. ** *p* < 0.01; ANOVA. (**C**,**D**) Western blot detected the expression of P53, Bax, and Bcl2 in HepG2 and MHCC-97H cells after CCl_4_ or MB interference, respectively. Data are presented as the mean ± SD. ** *p* < 0.01; ANOVA.

**Figure 8 molecules-28-07989-f008:**
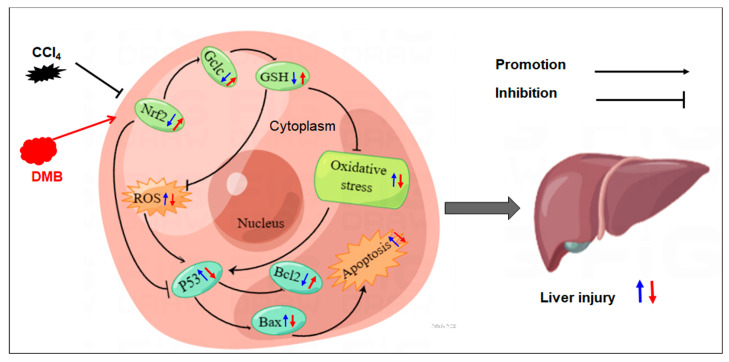
Proposed model by which DMB reverses the roles of CCl_4_-induced liver injury. Blue arrows, the effects of CCl_4_ on gene expression, oxidative stress, cell apoptosis, and liver injury. Red arrows, the effects of DMB on ameliorating the roles of CCl_4_-induced liver injury.

## Data Availability

All of the data are contained within the article and the Appendix A.

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
