# Peer review of "Dimethyl Bisphenolate Ameliorates Carbon Tetrachloride-Induced Liver Injury by Regulating Oxidative Stress-Related Genes"

_molecules, 2023, doi:10.3390/molecules28247989_

Round 1

Reviewer 1 Report

Comments and Suggestions for Authors

Dear authors,

Upon careful investigation of your manuscript, I find its content and scientific novelty intriguing. The respected authors have critically followed a strict experimental strategy to evaluate their idea, and the results are interesting. However, there are several points that should be addressed before publishing this manuscript. Please refer to the following comments and provide detailed answers to each query. Therefore, revise the manuscript content accordingly.

1- Please check the grammatical errors of your paper text and revise the manuscript accordingly.

2- Please check your reference list and make sure are cited references are available in the literature. If the respected authors inadvertently cited problematic papers (such as retracted papers or papers that are criticized in the PubPeer database), remove these references (if any) and refer to validated and original studies.

3- If there are data that should be supplemented, please kindly supplement all essential data for the academic readers of this paper. This helps you to increase the reproducibility of your experimental results and increase the quality of your work.

4- Please carefully check the authors’ affiliations, email addresses, and the order of authors in the authors’ list to prevent further corrections that might be needed to amend the manuscript after publishing.

5- One of the most professional aspects of scientific publishing is providing a succinct flowchart for all experimental methods that have been conducted during research. In this regard, I highly recommend you provide a graphical abstract for this paper to show how the experimental methods were conducted in this research. If you are not allowed to add a graphical abstract within the paper based on the journal style-you can instead supplement it. Indeed, please provide a flowchart for the M&M section of this paper.

6- Please add the Catalogue number of all chemicals used in this study. The authors can list all applied chemicals in a supplementary file and add their Cat. Number and manufacturer name to that table.

7- Please add global statistics on liver diseases to the introduction section. You can use WHO statistics or global trends that are specifically available for each disease.

8- According to my experience, emails that are hosted on the 163.com website are not reachable. Please use your academic email or use well-known email provider servers to help academic readers who want to communicate with the corresponding author.

9- Please provide a list of abbreviations for all summarized terms in the manuscript.

10- Lines 38-41: Add a reference to this section.

11- In the introduction section, the respected authors only mentioned that Pelargonidin can affect the expression of the Nrf2 transcription factor. In this section, the authors can generally discuss the benefits of phenolic compounds in the regulation of Nrf2 protein.

12- Figure 5 sections G/H details are not transparent. Please improve the resolution of these figures.

13- Please add a concluding remark at the end of the discussion and briefly summarize the top findings of this study.

14- Please cite all protocols and procedures that were used to conduct experimental assays of this study.

15- Lines 400-404: Please insert this section at the end of the discussion.

16- Please add an ethical approval number for using animals in this study.

17- Please add DOI identifiers to all cited literature in this study.

18- Line 84: Delete “.” Before reference [11,12].

19- Line 85: Based on which protocol or rationale the studied doses were selected for this study? Please clarify this case if the respected authors use the specific methodology to select these concentrations for primary injections in this study.

20- While the respected authors focused on DMB in this study, in one of their control groups, they used silymarin. Why do the esteemed authors use this phenolic compound? If the study tries to highlight the benefits of specific phenolic compounds in addition to synthesized DMB, the respected authors should address that.

21- The authors mentioned that the administration of DMB could significantly affect the level of antioxidant enzymes. Because these enzymes are involved in the progression of inflammation, the respected authors should show how DMB can affect the expression of inflammatory factors before/after administration of the synthesized compound. Please clarify this case and discuss the results in detail in your discussion section.

22- Why do the respected authors only focus on the serum level of the studied enzymes? Is it not better you measure their expression level after administration of your synthesized compound?

23- Why do the respected authors use the Kruskal–Wallis test instead of the one-way ANOVA?  In the majority of statistical analyses used to evaluate biological data, the researchers mainly used parametric procedures to analyze their data. Using the Kruskal–Wallis test means that any distributional assumption was not considered in this study. Please check your statistical analysis and make sure that the correct statistical method is used for comparing the obtained experimental data.

24- Line 145: Please correct the typography of “abovementioned” to “the above-mentioned”.  Please do this for the rest of the manuscript text.

25- Lines 144-158: it is better for the respected authors to add the additional methodological explanations to the M&M section and in this section only focus on the results of their bioinformatics procedure.

25- How do the respected authors identify critical catalytic and binding residues of the studied active site? If you used the specific database to predict the active site of this protein, the name of that database should be added to the M&M section.

26- The respected authors mentioned that they used Swisstargetprediction, SEA, and SuperPhred, to find the major target of the synthesized compound. Based on which criteria the respected authors selected Nrf2 as the ultimate target of this compound. As I checked your compound SMILES file in these three databases, I found that it can also interact with other proteins. Can you please supplement the results of your searches in the above-mentioned databases and highlight this case how you selected the studied protein as a target of DMB that was used in your docking simulation section? Please also supplement the exact SMILEs file that you used to search these databases.

27- The respected authors used silymarin as positive control but experimental data related to this compound were not found within the paper and the authors briefly discussed its protective effects at the end of the discussion by referring to the previously published literature on this compound. Please compare the silymarin data with DMB and show how coupling these two compounds can enhance the protective effects in the liver.

28- More importantly, the blotted figures that were represented in the content of the paper were not discussed in detail. Please clarify about blot images and briefly discuss the differences between the given blot spots.

After critical reviewing of this paper, I find its structure to be good, and the respected authors have provided novel findings for the academic readers of this journal. After amending the manuscript based on the given comments, it can be considered for publication. Please carefully review the above-mentioned comments and provide clear answers to each query while revising the manuscript. After reviewing the final version of your paper, I will be able to judge the content and express my final decision. At this stage, my recommendation for this paper is "major revision."

Comments on the Quality of English Language

The English language is fine, but some typographical errors should be amended during the revision. 

Author Response

Dear Reviewer

Many thanks for your comments “Upon careful investigation of your manuscript, I find its content and scientific novelty intriguing. The respected authors have critically followed a strict experimental strategy to evaluate their idea, and the results are interesting.”. We have revised the manuscript and marked the new changes in the marked version. The point-by-point responses to the comments are shown as follows.

Question 1: Please check the grammatical errors of your paper text and revise the manuscript accordingly.

Response: we checked the manuscript carefully, and revised the grammatical errors.

Question 2: Please check your reference list and make sure are cited references are available in the literature. If the respected authors inadvertently cited problematic papers (such as retracted papers or papers that are criticized in the PubPeer database), remove these references (if any) and refer to validated and original studies.

Response: we checked the references carefully, and the cited references are available in the literature. No problematic papers are found in the references.

Question 3: If there are data that should be supplemented, please kindly supplement all essential data for the academic readers of this paper. This helps you to increase the reproducibility of your experimental results and increase the quality of your work.

Response: we provided all essential data, including some results data and methods in supplemental data.

Question 4: Please carefully check the authors’ affiliations, email addresses, and the order of authors in the authors’ list to prevent further corrections that might be needed to amend the manuscript after publishing.

Response: we carefully checked the author’s affiliations, email addresses, and the order of authors. The names, affiliations, and the order are correct. We replaced 163.com email as follows:

Correspondence: [email protected]; [email protected]; Department of Biochemistry and Molecular Biology, Binzhou Medical University, YanTai, Shandong, 264003, P.R.China

Question 5: One of the most professional aspects of scientific publishing is providing a succinct flowchart for all experimental methods that have been conducted during research. In this regard, I highly recommend you provide a graphical abstract for this paper to show how the experimental methods were conducted in this research. If you are not allowed to add a graphical abstract within the paper based on the journal style-you can instead supplement it. Indeed, please provide a flowchart for the M&M section of this paper.

Response: we provided a graphical abstract for this paper to show the experimental methods.

Question 6: Please add the Catalogue number of all chemicals used in this study. The authors can list all applied chemicals in a supplementary file and add their Cat. Number and manufacturer name to that table.

Response: we added the Catalogue number of all chemicals used in this manuscript.

Question 7: Please add global statistics on liver diseases to the introduction section. You can use WHO statistics or global trends that are specifically available for each disease.

Response: we added the data of WHO in the introduction as follows: “Hepatitis induces liver damage or cancer, which causes a million people death annually. WHO warns that the disease induced by viral hepatitis would kill more people than tuberculosis, malaria, and HIV combined by 2040, if the infection trends continue” 

Question 8: According to my experience, emails that are hosted on the 163.com website are not reachable. Please use your academic email or use well-known email provider servers to help academic readers who want to communicate with the corresponding author.

Response: we replaced the 163.com email as follows:

Correspondence: [email protected]; [email protected]; Department of Biochemistry and Molecular Biology, Binzhou Medical University, YanTai, Shandong, 264003, P.R.China

Question 9: Please provide a list of abbreviations for all summarized terms in the manuscript.

Response: we provided a list of abbreviations at the end of the manuscript.

Question 10: Lines 38-41: Add a reference to this section.

Response: we provided the reference: Katarey, D.; Verma, S.Drug-induced liver injury. Clinical medicine (London, England)  2016, 16(Suppl 6), s104–s109. https://doi.org/10.7861/clinmedicine.16-6-s104.

Question 11: In the introduction section, the respected authors only mentioned that Pelargonidin can affect the expression of the Nrf2 transcription factor. In this section, the authors can generally discuss the benefits of phenolic compounds in the regulation of Nrf2 protein.

Response: we discussed in this section as follows: “Various phenolic compounds can regulate nuclear factor-κB or Nrf-2 to exert important roles in antioxidant response [11]. It is known that phenolic compounds show an-ti-inflammatory properties to treat rheumatoid arthritis or inflammatory bowel disease [11].”

Question 12: Figure 5 sections G/H details are not transparent. Please improve the resolution of these figures.

Response: we improved the resolution of these figures, and provided the Figure-5-n.

Question 13: Please add a concluding remark at the end of the discussion and briefly summarize the top findings of this study.

Response: we added a concluding remark at the end of the discussion and summarized the top finding.

Question 14: Please cite all protocols and procedures that were used to conduct experimental assays of this study.

Response: Thank you for your comments. First, we provided more details on the methodology to enhance the paper's scientific rigor. Second, we supplemented the detailed method in supplemental data.

Question 15: Lines 400-404: Please insert this section at the end of the discussion.

Response: thank you for your comments. we inserted this section at the end of the discussion.

Question 16: Please add an ethical approval number for using animals in this study.

Response: thank you for your comments. we added in the Materials and Methods: “All animal experiments were approved by the Animal Ethics Committee of Binzhou Medical University (No. 2021-06-03).”

Question 17: Please add DOI identifiers to all cited literature in this study.

Response: we added the DOI identifiers to all cited references.

Question 18: Line 84: Delete “.” Before reference [11,12].

Response: we deleted “.” Before reference [11,12].

Question 19:  Line 85: Based on which protocol or rationale the studied doses were selected for this study? Please clarify this case if the respected authors use the specific methodology to select these concentrations for primary injections in this study.

Response: we set up different concentration of CCl4 based on previous report. In the text, we described: Based on the concentration of CCl4 in previous report [13], to further determine the appropriate dosage needed in this study, five different injection doses, namely 0, 300, 600, 900, 1200 μL/kg, were designed to treat SD rats. One day after a single injection, the blood of rats was collected from the apex of the heart and the serum was tested.

  1. Di Paola, R.; Modafferi, S.; Siracusa, R.; Cordaro, M.; D'Amico, R.; Ontario, M. L.; Interdonato, L.; Salinaro, A. T.; Fusco, R.; Impellizzeri, D.; et al. S-Acetyl-Glutathione Attenuates Carbon Tetrachloride-Induced Liver Injury by Modulating Oxidative Imbalance and Inflammation. International journal of molecular sciences 2022, 23, 4429. https://doi.org/10.3390/ijms23084429.

Question 20: While the respected authors focused on DMB in this study, in one of their control groups, they used silymarin. Why do the esteemed authors use this phenolic compound? If the study tries to highlight the benefits of specific phenolic compounds in addition to synthesized DMB, the respected authors should address that.

Response: we stated the reason of using silymarin in the discussion as follows: “Considering its roles in protecting liver injury through antioxidant and anti-apoptotic activities, silymarin was used as a positive control in this study. Interestingly, our results indicated that the protective effects of DMB on ameliorating liver injury were similar to or better than silymarin did.”

Question 21: The authors mentioned that the administration of DMB could significantly affect the level of antioxidant enzymes. Because these enzymes are involved in the progression of inflammation, the respected authors should show how DMB can affect the expression of inflammatory factors before/after administration of the synthesized compound. Please clarify this case and discuss the results in detail in your discussion section.

Response: we stated in discussion: “Here, we found that the new compound DMB significantly increased the content of GSH, Nrf2 and GCLC in the liver tissues of CCL4-induced liver injury of rats. Considering the roles of ROS in inflammation [26], our results indicated that DMB may ameliorate the liver inflammation through regulating GSH, Nrf2 and GCLC to reduce ROS.” 

Question 22: Why do the respected authors only focus on the serum level of the studied enzymes? Is it not better you measure their expression level after administration of your synthesized compound?

Response: The alanine aminotransfetase (ALT), aspartate aminotransfetase (AST), direct bilirubin (DBIL), total bilirubin (TBIL), alkaline phosphatase (ALP), and lactate dehydrogenase (LDH) in the serum are most common liver function indices to estimate liver injury in clinic. Therefore, we detected these enzymes expression in serum. After treatment of DMB, we detected Nrf2, GCLC, p53, Bax, and Bcl2 levels in vitro and in vivo. If it is necessary, we will detect these enzymes in the future.

Question 23: Why do the respected authors use the Kruskal–Wallis test instead of the one-way ANOVA?  In the majority of statistical analyses used to evaluate biological data, the researchers mainly used parametric procedures to analyze their data. Using the Kruskal–Wallis test means that any distributional assumption was not considered in this study. Please check your statistical analysis and make sure that the correct statistical method is used for comparing the obtained experimental data.

Response: The median (interquartile range) was present for abnormally distributed data, and the multiple groups were compared by Kruskal–Wallis H test. Normally distributed data were present as mean ± SD. The multiple groups were compared by ANOVA. Tukey test was used for multiple comparisons.

We stated the data is abnormally distributed data in the figure legends.

Question 24: Line 145: Please correct the typography of “abovementioned” to “the above-mentioned”.  Please do this for the rest of the manuscript text.

Response: We corrected the typography of “abovementioned” to “the above-mentioned”.

Question 25: Lines 144-158: it is better for the respected authors to add the additional methodological explanations to the M&M section and in this section only focus on the results of their bioinformatics procedure.

Response: We added the additional methodological explanations to the Materials and Methods, and focused on the results in the Results.

Question 26: 25- How do the respected authors identify critical catalytic and binding residues of the studied active site? If you used the specific database to predict the active site of this protein, the name of that database should be added to the M&M section.

Response: we used AutoDockTools software for molecular simulation docking, and described in the M&M section.

Question 27: The respected authors mentioned that they used Swisstargetprediction, SEA, and SuperPhred, to find the major target of the synthesized compound. Based on which criteria the respected authors selected Nrf2 as the ultimate target of this compound. As I checked your compound SMILES file in these three databases, I found that it can also interact with other proteins. Can you please supplement the results of your searches in the above-mentioned databases and highlight this case how you selected the studied protein as a target of DMB that was used in your docking simulation section? Please also supplement the exact SMILEs file that you used to search these databases.

Response: we provided the targets of SEA, and SuperPhred in supplemental Table 1, 2, and file of SMILES in supplemental data. In the results, we explained the reason to choose Nrf2 in this study:

The above-mentioned results showed that DMB ameliorated the CCl4-induced liver injury. To further investigate the molecular mechanism of DMB in liver injury, the possible binding proteins were then predicted by SEA and SuperPred, and results demonstrated that Nrf2 was a target of DMB (supplemental Table 1, 2). We next showed that arginine (ARG) at position 322, valine (VAL) at position 420, glycine (GLY) at position 511, and valine at position 512 in NRF2 can bind to DMB through hydrogen bond (Figure4A).

These results indicate that Nrf2 is a target of DMB. Nrf2 functions as a ubiquitous and evolutionarily conserved protein to counteract oxidative stress [21], which participates in the pathological process of liver injury [22]. Therefore, the roles of Nrf2 in liver injury were investigated in this study.

Question 28: The respected authors used silymarin as positive control but experimental data related to this compound were not found within the paper and the authors briefly discussed its protective effects at the end of the discussion by referring to the previously published literature on this compound. Please compare the silymarin data with DMB and show how coupling these two compounds can enhance the protective effects in the liver.

Response: That is a good question. In fact, the results of silymarin-treated group were in Figure 3 and Figure 4. We further described the results in the “Results” as follows: “The protective effects of DMB on biochemical indicators were similar as or better than43.75mg/Kg silymarin did (clinical drug as positive control treatment, Figure 3 A-F). HE staining further supported that the low, medium, and high doses of DMB amelio-rated the histological liver injury induced by CCl4 as silymarin did (Figure3G). Similar results of GSH and MDA were found in silymarin-treated rats (Figure 4B,C). DMB treatment decreased the expression of p53 and Bax but increased Bcl2 expression, and the changes of p53, Bax or Bcl2 induced by DMB is similar to the effects of silymarin on the expression of these genes (Figure 4D,G,H).”

In discussion, we stated that: “Considering its roles in protecting liver injury through antioxidant and anti-apoptotic activities, silymarin was used as a positive control in this study. Interestingly, our results indicated that the protective effects of DMB on ameliorating liver injury were similar to or better than silymarin did. These results indicate that DMB has potential application or combinates with silymarin in protection of liver injury.”

Question 29: More importantly, the blotted figures that were represented in the content of the paper were not discussed in detail. Please clarify about blot images and briefly discuss the differences between the given blot spots.

Response: in discussion we stated that: “Western blot showed that p53 and Bax were increased, but Nrf2 and Bcl2 were in CCl4-induced liver cells or tissues. Nevertheless, DMB treatment decreased the expression of P53 and Bax but increased Nrf2 and Bcl2 expression compared with CCl4-treated cells or tissues.”

Question 30: The English language is fine, but some typographical errors should be amended during the revision. 

Response: Thank you for your comments very much. We revised the manuscript carefully and corrected some typographical errors.

Best wishes.

Sincerely,

Shu-Yang Xie, PhD. Professor.

Department of Biochemistry and Molecular Biology

Binzhou Medical University

Laishan District, No346.

YanTai, ShanDong 264003

P.R. China

Phone: +86 535 6913070

Fax: +86 535 6913163

Reviewer 2 Report

Comments and Suggestions for Authors

The authors elucidated that dimethyl bisphenolate (DMB) has protective effects on carbon tetrachloride (CCl4)-induced liver injury through regulating oxidative stress-related genes. However, the mechanism underlying it has not been examined appropriately and their conclusion is not sufficiently supported by the experimental data.

Major comments

1. It is not clear how the concentrations of DMB were selected for this study, in general, based on previous results is not enough explanation. In cell systems cytotoxicity assessment should be based on the rules of ISO 10993-5:2009. The toxic concentration of the compound in animals is not known. Taking that all into consideration, the conclusion on the activity of the compound cannot be fully made.

2. In vivo study, histopathogic data of liver should be added. Qualified pathologist should observe the liver tissues by blind test.

3. HepG2 and MHCC-97H cells are transformed hepatoma cell lines. Please provide the rational why the authors used such transformed cell lines, not primary hepatocytes.

4. Authors should clarify which post-hoc test was used in the One way ANOVA.

5. It was disappointing that the authors did not attempt to provide any explanation of the mechanism by which DMB exerts its effect. As presented, the focus was solely on the effect of the chemical on various endpoints and it remains to be determined whether the effects are primary or secondary effects (i.e. does DMB produce this effect directly). Also, is the effect produced by DMB or its metabolite?

6. In the Discussion, It should be commented, if the used concentration of DMB can be reached in the body and therefore be of physiological relevance.

Comments on the Quality of English Language

The authors should provide the abbreviation list, and take care of numerous grammatical mistakes throughout the manuscript which detract from its quality.

Author Response

Dear Reviewer

Many thanks for your comments. We have revised the manuscript and marked the new changes in the marked version. The point-by-point responses to the comments are shown as follows.

Question 1: It is not clear how the concentrations of DMB were selected for this study, in general, based on previous results is not enough explanation. In cell systems cytotoxicity assessment should be based on the rules of ISO 10993-5:2009. The toxic concentration of the compound in animals is not known. Taking that all into consideration, the conclusion on the activity of the compound cannot be fully made.

Response: Thank you for your comment. The above-mentioned results indicated that DMB ameliorates the roles of CCl4-induced liver injury through regulating Nrf2-related genes. Next, we did experiment to assess the cytotoxicity of DMB in vitro and in vivo. Results showed that 10 μg/mL, 20 μg/mL, 30 μg/mL, 60 μg/mL and 100 μg/mL of DMB didn’t have cytotoxicity to MHCC-97H and 293T cells (supplemental Figure 1). In vitro study, treatment of 45 mg/kg of DMB did not induce significant changes of liver tissues (supplemental Figure 2) through histological analysis of liver tissues. Results supported that DMB did not significantly induce cytotoxicity in vitro and in vivo.

In discussion, we also stated that: Our finding indicated that DMB might be applicated in clinical practice for liver injury in the future. However, the detailed regulating mechanism and the cytotoxicity assessment of DMB, such as side effects and concentration in vivo, must be further investigated.

Question 2: In vivo study, histopathogic data of liver should be added. Qualified pathologist should observe the liver tissues by blind test.

Response: First, The SD rats were randomly divided into normal control group, model group, silymarin group (43.75mg/kg based on clinical medication conversion), and DMB low (15 mg/kg), medium (30mg/kg), and high dose groups (45mg/kg), with 6 rats each group by a "blinded" investigator. Second, The pathological changes of rat liver tissues were observed under a microscope by a liver pathologist. Third, the changes of ALT, AST, DBIL, TBIL, ALP, and LDH in the serum of rats, as well as Pathological changes of liver tissues were analyzed in the manuscript. These data were shown in Figure 2 and 3.

Question 3: HepG2 and MHCC-97H cells are transformed hepatoma cell lines. Please provide the rational why the authors used such transformed cell lines, not primary hepatocytes.

Response: That is a good suggestion. HepG2 and MHCC-97H cells are transformed hepatoma cell lines. They are easy to be cultured to reflect and study the changes of liver cells. We will study the effects of DMB on primary culture of liver cells in the following experiments.

Question 4: Authors should clarify which post-hoc test was used in the One way ANOVA.

Response: thank you for your comments. Tukey test was used for multiple comparisons, and we revised in the “Materials and Methods”.

Question 5: It was disappointing that the authors did not attempt to provide any explanation of the mechanism by which DMB exerts its effect. As presented, the focus was solely on the effect of the chemical on various endpoints and it remains to be determined whether the effects are primary or secondary effects (i.e. does DMB produce this effect directly). Also, is the effect produced by DMB or its metabolite?

Response: In the discussion, we explained the possible mechanism by which DMB exerts its effect based on our results. We also discussed the possible application of DMB in liver injury.

Question 6: In the Discussion, It should be commented, if the used concentration of DMB can be reached in the body and therefore be of physiological relevance.

Response: in the discussion, we discussed as follows: “Our finding indicated that DMB might be applicated in clinical practice for liver injury in the future. However, the detailed regulating mechanism and the cytotoxicity assessment of DMB, such as side effects and concentration in vivo, must be further investigated.”

Question 7: The authors should provide the abbreviation list, and take care of numerous grammatical mistakes throughout the manuscript which detract from its quality.

Response: Thank you for your comments. We provided the abbreviation list at the end of the manuscript, and revised the numerous grammatical mistakes throughout the manuscript.

Best wishes.

Sincerely,

Shu-Yang Xie, PhD. Professor.

Department of Biochemistry and Molecular Biology

Binzhou Medical University

Laishan District, No346.

YanTai, ShanDong 264003

P.R. China

Phone: +86 535 6913070

Fax: +86 535 6913163

Reviewer 3 Report

Comments and Suggestions for Authors

The author presents a thorough investigation into the novel compound DMB and its potential in alleviating CCl4-induced liver injury by modulating oxidative stress. Confirming DMB's structure through hydrogen spectrum and mass spectrometry provides a strong basis, yet providing more detailed information on these methods would bolster the paper's credibility. The study offers compelling evidence of DMB's protective impact on liver injury via diverse biochemical and molecular indicators. Notably, the reductions in ALT, AST, DBIL, TBIL, ALP, and LDH in CCl4-treated rats, akin to silymarin's effects, are significant. Furthermore, the changes in ROS levels, markers of oxidative stress (GSH, Nrf2, GCLC, MDA), and modulation of P53, Bax, and Bcl2 expression endorse DMB's protective role.

Enhancing the paper's scientific rigor would involve providing more details on the methodology used to measure the mentioned biochemical indicators and conduct molecular analyses. Additionally, clarifying the precise mechanisms by which DMB regulates oxidative stress-related genes would enrich the study's depth and potential for translation. Addressing potential limitations or challenges, such as side effects, would offer a more comprehensive understanding of DMB's efficacy and safety profile.

Highlighting the potential implications of these findings for clinical applications or future research pathways would enhance the paper's impact. Overall, the manuscript holds promise regarding DMB's protective effects against CCl4-induced liver injury. Detailed methodologies and elucidation of underlying mechanisms would significantly contribute to its scientific relevance and translational potential.

The following issues need to be corrected before publishing the manuscript:  The 13C-NMR of the DMB compound needs to be presented. The compound's purity should be stated. Clarification is required to ensure that the proton (1H-NMR) counting matches the structure.

Comments on the Quality of English Language

The writing is clear, precise, and effectively communicates the research findings. Minor revisions for grammar and syntax could further enhance the readability and overall quality of the manuscript. 

Author Response

Dear Reviewer,

Many thanks for reviewing our manuscript. We have revised the manuscript and marked the new changes in the marked version. The point-by-point responses to the comments are shown as follows.

Question 1: Enhancing the paper's scientific rigor would involve providing more details on the methodology used to measure the mentioned biochemical indicators and conduct molecular analyses. Additionally, clarifying the precise mechanisms by which DMB regulates oxidative stress-related genes would enrich the study's depth and potential for translation. Addressing potential limitations or challenges, such as side effects, would offer a more comprehensive understanding of DMB's efficacy and safety profile.

Response: Thank you for your comments. First, we provided more details on the methodology to enhance the paper's scientific rigor. Second, in the discussion, we added the possible mechanisms by which DMB regulates oxidative stress-related genes, and addressed the potential limitations or challenges, such as side effects.

Question 2: Highlighting the potential implications of these findings for clinical applications or future research pathways would enhance the paper's impact. Overall, the manuscript holds promise regarding DMB's protective effects against CCl4-induced liver injury. Detailed methodologies and elucidation of underlying mechanisms would significantly contribute to its scientific relevance and translational potential.

Response: Thank you for your comments. In the discussion, we highlighted the potential implications of these findings for clinical applications or future research pathways. We also provided more details on the methodology and the possible mechanisms of DMB's protective effects against CCl4-induced liver injury.

Question 3: The following issues need to be corrected before publishing the manuscript:  The 13C-NMR of the DMB compound needs to be presented. The compound's purity should be stated. Clarification is required to ensure that the proton (1H-NMR) counting matches the structure.

Response: The compound’s purity was analyzed by HPLC, and 1H-NMR were displayed in the manuscript as follows.

After the reaction, a total of 10.4g yellow powder was obtained through purification by HPLC, with a yield rate of 96.3% and a purity of 99.3%.

The results of 1H-NMR hydrogen spectrum as follows: 1H NMR(500MHz, Acetone-d6): δ3.63(s, 3H), 3.81(s, 3H), 6.34(d, J=16.41Hz, 1H) , 7.86(d, J=16.41, 1H), 8.21(s, -OH), 6.82(s, -OH), 6.81(dd, J=9.47Hz, J=1.93Hz, 1H), 6.57(d, J=9.47Hz, 1H), 6.88(d, J=1.93Hz, 1H), 7.58(s, 1H), 6.41(s, -OH), 7.22(s, -OH), 7.23(d, J=9.26Hz, 1H), 6.63(d, J=9.26Hz, 1H).

These 13C-NMR analysis will take a relatively long time (more than 10 days). We must complete the revision in 10 days. If it is necessary, we will provide in the next month.

Question 4: The writing is clear, precise, and effectively communicates the research findings. Minor revisions for grammar and syntax could further enhance the readability and overall quality of the manuscript. 

Response: Thank you for your comments. We revised the grammar and syntax carefully in the manuscript.

Best wishes.

Sincerely,

Shu-Yang Xie, PhD. Professor.

Department of Biochemistry and Molecular Biology

Binzhou Medical University

Laishan District, No346.

YanTai, ShanDong 264003

P.R. China

Phone: +86 535 6913070

Fax: +86 535 6913163

Round 2

Reviewer 1 Report

Comments and Suggestions for Authors

Accept. 

Reviewer 2 Report

Comments and Suggestions for Authors

In this revised vesrion of the paper, the authous have properly replyed the comments.